# Prevalence of *Listeria monocytogenes* in RTE Meat Products of Quevedo (Ecuador)

**DOI:** 10.3390/foods12152956

**Published:** 2023-08-04

**Authors:** Gary Alex Meza-Bone, Jessica Sayonara Meza Bone, Ángel Cedeño, Irene Martín, Alberto Martín, Naga Raju Maddela, Juan J. Córdoba

**Affiliations:** 1Ruminology Laboratory, Faculty of Animal and Biological Sciences, State Technical University of Quevedo, Quevedo 120301, Ecuador; gmeza@uteq.edu.ec; 2Instituto Superior Tecnológico Ciudad de Valencia, Los Ríos, Quevedo 948196, Ecuador; 3Biotechnology Laboratory, Microbiology, Science and Technology Research Department, State Technical University of Quevedo, Quevedo 120301, Ecuador; acedenom@uteq.edu.ec; 4Higiene y Seguridad Alimentaria, Instituto Universitario de Investigación de Carne y Productos Cárnicos (IProCar), Facultad de Veterinaria, Universidad de Extremadura, 10003 Cáceres, Spain; iremartint@unex.es (I.M.); jcordoba@unex.es (J.J.C.); 5Department of Biological Sciences, Faculty of Health Sciences, Technical University of Manabí, Portoviejo 130103, Ecuador; raju.maddela@utm.edu.ec

**Keywords:** *Listeria monocytogenes*, meat products, Quevedo, Ecuador, foodborne infection

## Abstract

*Listeria monocytogenes* is a foodborne pathogen that causes listeriosis and can be a problem in areas where meat products are sold at unregulated storage temperatures. In this work, the prevalence of *L. monocytogenes* was determined in the five most widely traded meat products in the province of Quevedo (Ecuador): bacon, “chorizo paisa”, grilled hamburger meat, mortadella, and salami. A total of 1000 samples of these products were analyzed in two seasons of the year (dry season/rainy season). All *L. monocytogenes* isolates were confirmed by PCR with primers designed for the *iap* gene. Furthermore, the positive samples were quantified for *L. monocytogenes*. Of the 1000 meat products analyzed, 163 were positive for *L. monocytogenes* (16.3%). The prevalence of *L. monocytogenes* in the two seasons in different meat products was as follows: 22.5% in mortadella, 19% in hamburger meat, 15% in bacon, 14.5% in chorizo paisa and 10.5% in salami. In addition, the concentration of *L. monocytogenes* in most of the positive samples was in the range of 4–6 log CFU/g or even higher. The results show the need for improvements in the hygienic measures and meat storage temperatures in Quevedo (Ecuador) to avoid risks of foodborne listeriosis.

## 1. Introduction

In recent years, the consumption of ready-to-eat foods (RTE) has increased due to changes in lifestyle [1]. RTE meat products represent a high risk of microbial contamination, allowing for the development of foodborne pathogens and posing a serious threat to public health [2].

The presence of *Listeria monocytogenes (L. monocytogenes)* in RTE meat products is a threat to public health. This bacterium is linked to primary production livestock [3], feedlots, slaughter point, transportation, and retail [4,5,6]. In addition, *L. monocytogenes* is characterized as a halotolerant bacterium, and influences food safety under different conditions, such as poorly sanitized areas, osmotic pressure, a low pH, and refrigeration temperatures [7].

Listeriosis is an infection caused by the consumption of foods contaminated by *L. monocytogenes,* a bacillus that is the subject of several studies in the meat industry [8,9,10]. The incidence of the disease in the population is low, but it presents high morbidity and mortality, and a hospitalization rate of more than 95% [11,12]. There is a high prevalence of listeriosis-based health risk in the following groups: elderly persons, pregnant women, fetuses, immunocompromised people (such as those with acquired immunodeficiency syndrome (AIDS)), cancer patients, and persons those who have undergone organ transplantation [13,14].

The consumption of contaminated foods with *L. monocytogenes,* such as turkey sausage, was found to cause listeriosis [15]. The vast majority of meat products have been involved in listeriosis outbreaks or sporadic cases. Vacuum-packed and processed meat products sold in stores or supermarkets that have not been subjected to temperature control that experience an increase in the storage temperature according to the season of the year [16] could cause listeriosis. In fact, more than 75% of meat contamination in stores was attributed to *L. monocytogenes* [17]. The incidence of meat contamination by *L. monocytogenes* in different countries is as follows: 36.73% in Nigeria [18], 1.7% in Japan [19], 0.1% in Poland [20], 3.1% in Chile [21].

In Ecuador, the impact of listeriosis on public health is not well-known because it is not a notifiable disease, despite the severity of its symptoms. *L. monocytogenes* has been detected by microbiological and molecular methods, both in commercialized foods and on the surface of equipment and utensils that are commercialized in this country [22]. However, the prevalence of this foodborne pathogen in meat products traded in Ecuador has not been established. A study of the prevalence of *L. monocytogenes* in meat products would be of relevance because meat products are traded at unregulated temperatures, and experience fluctuations in storage temperature depending on the season of year. Therefore, it is necessary to establish the prevalence of this pathogenic bacterium in meat products in Ecuador, establishing a relationship with the different seasons of the year. The objective of this study was to determine the prevalence of *L. monocytogenes* in the most widely traded meat products in Quevedo (Ecuador), considering two different climatic seasons of the year.

## 2. Materials and Methods

### 2.1. Sampling

To accomplish this study, different RTE meat products that are widely marketed in Quevedo (Ecuador) were purchased between January 2021 and January 2022 (Table 1). All samples were made according to the Mandatory Ecuadorian Technical Standard (NTE INEN 1338-2102). A description of these products is as follow: salami is a dry-cured fermented sausage made from pork and/or bovine meat, with pork fat, salt, sugar, spices, and with or without the addition of liquor; “chorizo Paisa” sausage is a product stuffed in natural or artificial casings, ripened for a short period, and smoked or not; bacon is a product obtained from the abdominal wall (bacon) or from the subcutaneous adipose tissue of pigs, cooked and smoked; mortadella is an emulsified mass prepared with selected meat and fat from farm animals and other ingredients, and stuffed in natural or artificial casings and cooked; roasted hamburger meat is the ground (or minced) meat of animals, and is homogenized, preformed and precooked, with ingredients and additives for permitted use.

All meat products were purchased in the market and vacuum-packed under refrigerated conditions that vary between 2 °C and 14 °C in the dry season and from 4 °C and 14 °C in the rainy season. Five types of commercially available meat products were sampled (1000 in total; 200 per each type) and collected from three local supermarkets (each one placed in the south, central and north) from Quevedo city, as follows: A: salami (*n* = 200), B: “chorizo paisa” (*n* = 200), C: bacon (*n* = 200), D: roasted hamburger meat (*n* = 200), E: mortadella (*n* = 200) (Table 1). Samples were randomly selected on the day of purchase. All the samples were placed in a portable refrigerator, keeping the temperature below 4 °C during transport, and the tests began once they arrived at the laboratory.

### 2.2. Detection of Listeria monocytogenes in Meat Products

The detection of *L. monocytogenes* was carried out according to the ISO 11290, part 01 [23]. According to this method, 25 g of each meat product sample was homogenized with 225 mL of Half-Fraser TM MEDIA broth (Titan Biotech, Ltd., Rajasthan, India) and incubated for 24 h at 30 °C. Then, 0.1 mL of culture primary enrichment was transferred to 10 mL Listeria Fraser TM MEDIA broth (Titan Biotech, Ltd.) and incubated at 37 °C for 24 h. Characteristic colonies of *L. monocytogenes* on Listeria chromogenic agar (blue–green in color and surrounded by an opaque halo) were determined according to Ottaviani and Agosti 1997 [24].

### 2.3. Listeria monocytogenes Enumeration in Meat Products

The enumeration of *L. monocytogenes* took place in accordance with the recommendations of ISO 11290, part 02 [23]. For this, 25 g of samples were diluted in 225 mL of buffered peptone water and left for 1 h at 20 °C to allow for the revitalization of stressed cells of *L. monocytogenes*. Then, decimal dilutions were prepared, and 0.1 mL of each sample was transferred to Listeria chromogenic agar plates and incubated at 37 °C. After 24–48 h of incubation, the plates were examined, and the characteristic colonies of presumptive *L. monocytogenes* were counted in log colony-forming units (CFU)/g [23].

### 2.4. Identification of Listeria monocytogenes by Amplification of the Iap Gene

#### 2.4.1. Extraction of DNA

The DNA extraction of the suspected *L. monocytogenes* isolates took place with the Invitrogen brand commercial Kit (K1820-01) (Thermo Fisher Scientific, Waltham, MA, USA), following the manufacturer’s instructions for Gram-positive bacteria. The extraction and preparation of the DNA was achieved using pure cultures. The cell pellet was suspended in 180 μL of Invitrogen lysozyme digestion buffer (Thermo Fisher Scientific) (20 mg/mL) and incubated at 37 °C for 30 min. Subsequently, the purified DNA was stored at –20 °C until use.

#### 2.4.2. Identification of *Listeria monocytogenes* by PCR Assay

A total of 163 *L. monocytogenes* strains were identified by a DNA sequence analysis of the *iap* gene using a classical PCR system with Mono A and Lis 1B primers designed from the *iap* gene [25]. Primer pairs Mono A: 5′-CAAACTGCTAACACAGCTACT-3′ and Lis1B: 5′-TTATACGCGACCGAAGCCAAC-3′ were used to amplify a 660 bp region of the *iap* gene. The PCR was performed with the final volume (50 µL) of the reaction, which contained 29 µL of sterile ultrapure water, 15 µL of Dreamtaq green PCR master mix (Thermo Scientific), 2 µL of each of two primers (32.8 nmol of Mono A and 30.1 nmol of Lys 1B), and 2 µL of genomic DNA. The amplification conditions were slightly modified, as indicated below: initial DNA denaturation at 94 °C for 1 min, followed by 30 cycles of 10 min at 94 °C, annealing for 1 min at 55 °C, and 1 min at 72 °C, with an extension of the amplified product at 72 °C for 7 min. PCR was performed in the Applied Biosystems Thermal Cycler (Foster City, CA, USA).

The PCR products obtained were separated on 2% agarose using gel electrophoresis with 1x Tris Acetate EDTA buffer (Invitrogen) at 100 V for 40 min. Gels were stained by ethidium bromide (2 µg/mL) from Invitrogen and the 1 kb plus DNA Ladder from Invitrogen molecular weight marker was used to determine the size of the DNA fragments. DNA was visualized in an ultraviolet light transilluminator and then photographed using the Canon Power Shot G9 camera. Only a 660 bp amplicon (in the range of 100 to 1000 bp) was detected in the isolates that were confirmed as *L. monocytogenes*. 

### 2.5. Statistical Analysis

The statistical analysis was carried out using the IBM SPSS Statistics program and the Microsoft Excel Professional Plus 2016 software. For a statistical analysis of the data, both independent variables (prevalence of *L. monocytogenes*) and dependent variables (salami, chorizo paisa, bacon, hamburger meat, mortadella) were used. Once the dependent and independent variables of the analysis were determined, a normal distribution of the data obtained was studied using Pearson’s Chi-square test. Statistical significance was established at *p* ≤ 0.05.

## 3. Results

### 3.1. Detection of Listeria monocytogenes in RTE Meat Products

From the 1000 samples of tested meat products, 163 were positive for *L. monocytogenes* as primarily confirmed by the characteristic colonies of this pathogen. The PCR results of the *iap* gene provided a final confirmation of the identity of the presumptive isolates in this study, with an amplification of a 660 bp PCR product (Figure 1).

### 3.2. Prevalence of Listeria monocytogenes in RTE Meat Products

All the samples of the five RTE meat products analyzed in this study were evaluated based on the criteria proposed in the European Regulations. Table 2 shows the results of the analysis of the prevalence of *L. monocytogenes* in 1000 samples from five groups (of 200 samples) of meat products (salami, “chorizo paisa”, bacon, hamburger meat and mortadella). Results indicated that 10.5% of salami, 14.5% of “chorizo paisa”, 15% of bacon, 19% of hamburger meat and 22.5% of mortadella were positive for *L. monocytogenes* (Table 2). 

Considering the different seasons (dry and rainy), there was a higher prevalence of *L. monocytogenes* in the dry season (22.2%) than in the rainy season (10.4%) (Figure 2). 

In the same way, the results of the prevalence of *L. monocytogenes* were presented according to the sampling area. There was a higher prevalence of *L. monocytogenes* in the meat samples of the central area (40.5%) than in the southern (39.3%) and northern zones (20.2%), with temperatures fluctuating between 2 and 14 °C (Table 3).

### 3.3. Listeria monocytogenes Counts in Meat Products

The counts of *L. monocytogenes* according to the percentages of the different meat products are shown in Figure 3.

Most (90.9%) salami samples that were positive for *L. monocytogenes* showed levels of this pathogenic bacteria between 4 and 6 log CFU/g, while the remaining 9.1% of these samples presented counts higher than 6 log CFU/g. Similarly, 89.7% of “chorizo paisa” with *L. monocytogenes* showed counts of 4–6 log CFU/g, and 10.3% of the positive samples of this product had levels between 2 and 4 log CFU/g (Figure 3). With respect to smoked bacon, 56.7% of the samples that were positive for *L. monocytogenes* showed counts in the range of 2–4 log CFU/g, while 43.3% of them showed counts of 4–6 log CFU/g. In hamburger meat, more than 53% of the samples positive for *L. monocytogenes* had levels higher than 4 log CFU/g and more than 46.2% had counts greater than 6 log CFU/g. In the same way, of the samples of “mortadella” that were positive for *L. monocytogenes*, 26.1% presented counts between 4 and 6 log CFU/g, and the remaining samples (73.9%) showed levels greater than 6 log CFU/g (Figure 3). 

The results reported in this study confirmed the predominance of *L. monocytogenes*, as well as log CFU/g counts in the meat products, for the two seasons under study (Figure 4). The highest amount of log CFU/g of *L. monocytogenes* was found in the dry season (56.8%) and the rainy season (52.7%) was found to be between 4 and 6 log CFU/g; this indicates that the meat product samples presented high levels of contamination. The counts of higher than 6 log CFU/g in the two seasons presented almost equal values in the dry season (32.4%) and in the rainy season (32.7%), which indicated that the meat product samples presented levels of deterioration. The counts with levels of 2–4 log CFU/g were the lowest values recorded between the two studied seasons (Figure 4).

## 4. Discussion

The prevalence of *L. monocytogenes* in this study was found to be 22.2% in the dry season and 10.4% in the rainy season. Therefore, it is essential to follow safe food storage guidelines, refrigerating meat products at the proper temperature, to prevent the growth of pathogenic bacteria such as *L. monocytogenes.* According to the literature, the possible causes of meat contamination by *L. monocytogenes* were the storage of raw material in inadequate conditions, poor hygienic practices and deficiencies in the manufacturing (such as the processes, finishing, packaging, product storage, distribution, marketing storage and commercial refrigerators) [26,27]. This is consistent with the literature [28], which states that *L. monocytogenes* are present in the environment, farms, forage, animal feed, and also in the food-processing environment, including the industry and establishments for preparation, and food trade [29]. Several studies have documented the incidence of the pathogen, with prevalence levels reaching 40–45% [30].

In this research, incorrect storage temperatures in commercial refrigerators that fluctuated between 2 °C and 14 °C were reported. Such conditions favour the adaptation of growth and resistance of *L. monocytogenes* in meat products, contributing to the risk of food poisoning [31]. According to Iacumin et al. [32], inadequate temperature conditions during the storage of packaged cooked ham in supermarket refrigerators can allow for the growth of *L. monocytogenes,* reaching values capable of producing illnesses in the consumer. The growth of *L. monocytogenes* is favored by thermal abuse or the long shelf-life to which these products are subjected (e.g., from 23 to 30 days for sausages and up to 60 days for diced cooked ham). It is worth noting that, with the arrival of the warm season and the increase in environmental temperature by 25–32 °C, the level of this pathogenic bacterium rises. For example, an increase in the counts of *L. monocytogenes* from 2 to 4 logarithmic units was reported in vacuum-packed meat products stored at 4 °C [33]. Nothing the literature and the results of the present investigation, an exhaustive control of the temperature in meat products in Quevedo (Ecuador) is necessary to minimize the risk of the growth of *L. monocytogenes* during commercialization.

In this study, the presence of *L. monocytogenes* was observed in 163 samples of meat products at levels higher than 2 log CFU/g. According to the European Commission Regulation EC 2073/2005, the analyzed products are in the category of RTE foods that can support the growth of *L. monocytogenes*. Therefore, all these products have to comply with the criteria of the absence of *L. monocytogenes* in 25 g products before leaving the immediate control of the food business operator, and less than 2 log CFU/g counts of *L. monocytogenes* in the products placed on the market during their shelf-life. Thus, according to the above European Commission Regulation, non-compliance with the criteria regarding levels of *L. monocytogenes* was found in all the analyzed meat products in the present work. This fact highlights the relevance of improving preventive measures to avoid contamination and the proliferation of *L. monocytogenes* during meat products; preparation in the industry and throughout the distribution chain in refrigerated meat products.

The European Food Safety Authority (EFSA) evaluates the risk of contamination of RTE foods, considering that cases of listeriosis could be due to products found to be loaded with levels of more than 200 CFU/g of *L. monocytogenes*. The official United States of America regulatory policy for *L. monocytogenes* is zero-tolerance in RTE foods (FDA, Washington, DC, USA, 354 2003) that support growth and have a long shelf-life [34]. This suggests that there could be a higher risk of human infection by *L. monocytogenes* among the RTE meat products analyzed in the present work.

*L. monocytogenes* was found in 1.52–11.6% of the fermented meat products analyzed, although different conditions (e.g., sodium chloride, sodium nitrite and pH) were used to inhibit the growth of *L. monocytogenes* in fermented sausage [35,36]. This is in accordance with the results reported in different works [37,38], where *L. monocytogenes* was found in 14% of tested samples of traditionally dry and smoked fermented sausages. In addition, the results found in the present work agree with the other investigations, where the reported counts of *L. monocytogenes* were higher than 2 log CFU/g in dry-cured fermented “chorizo”, attributed mainly to the high levels of water activity during ripening (above the limit of growth for this pathogenic bacteria), resistance to curing salts and poor hygienic conditions in industrial production.

The prevalence of *L. monocytogenes* in the selected meat samples of Quevedo (Ecuador) analyzed in the present work were higher than those found by other authors in other parts of the world for RTE meat products. Uyttendaele et al. [39] found 4.9% of positive samples after analyzing 3405 cooked meat products. Elson et al. [40] detected a lower prevalence (2.2%) of *L. monocytogenes* in 4078 samples of cooked meat. This low prevalence was also found by Wong et al. [41], who analyzed 300 samples of pates and 104 of packaged cooked ham, where only 1% of samples were found to be positive *L. monocytogenes* and only 1 sample had a count of 3 log CFU/g. Two more recent studies have shown incidences of *L. monocytogenes* in these same products of 1.1%, with counts below 100 CFU/g [42] and 7.4% [43].

The tolerance levels of *L. monocytogenes* must be zero according to the policies of the United States government [44]. However, in the present work, for RTE products such as mortadella, more than 25% of the analyzed samples had counts of *L. monocytogenes* between 4–6 log CFU/g) and 73.90% (>6 log CFU/g). The International Commission on Microbiological Specifications for Food (ICMSF) set a limit of 2 log CFU/g for *L. monocytogenes* in RTE foods, which is the acceptable limit for low-risk consumers [45], since this pathogenic bacteria is frequently found in processed and RTE foods as well as in fermented meat products [46,47].

In the present work, the level of *L. monocytogenes* found in positive samples of fermented cured salami was 90.90% of 4–6 log CFU/g and 9.10% of >6 log CFU/g, probably due to possible recontamination during distribution or at the point of sale, as well as the water activity of the product and storage temperature, which allow for the growth of this pathogenic bacteria [48].

## 5. Conclusions

The prevalence of *Listeria monocytogenes* in meat products marketed in the Quevedo region (Ecuador) presented a higher percentage in the dry season, with hygienic measures during processing and refrigeration temperature being the most important risk factors in infection by the consumption of contaminated foods. The majority of *L.-monocytogenes*-positive samples in all tested products showed levels ranging from 4 to 6 log CFU/g or higher. It is essential to design programs that help prevent contamination or inhibit the growth of *L. monocytogenes* in meat products, good manufacturing practices, adequate cleaning, sanitation and hygiene programs, and effective temperature control throughout the chain. In addition, production, distribution and storage in the commercial refrigerators should guarantee the safety of meat products in supermarkets in Quevedo (Ecuador) to avoid the risk of infection with foodborne listeriosis.

## Figures and Tables

**Figure 1 foods-12-02956-f001:**
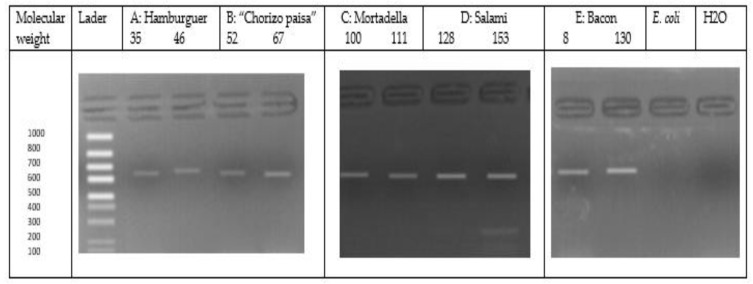
Amplified DNA fragments of the *iap* gene in suspected *L. monocytogenes* isolates obtained from the different meat products. Representative wells are depicted. (**A**) Isolate 35, 46, obtained from roasted hamburger meat; (**B**) Isolate 52, 67, from “chorizo paisa”; (**C**) Isolate 100, 111, obtained from mortadella; (**D**) Isolate 128, 153, obtained from salami; (**E**) Isolate 8, 130, obtained from bacon. In all the isolates, a PCR product of approximately 660 bp was found (MP. Ladder marker; *E. coli*. *Escherichia coli*—negative control; H_2_O negative control).

**Figure 2 foods-12-02956-f002:**
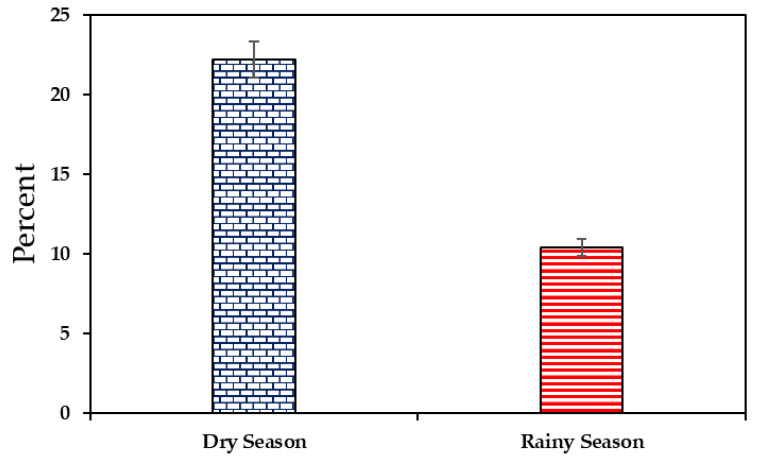
Prevalence of *Listeria monocytogenes* in meat products according to the season (dry or rainy).

**Figure 3 foods-12-02956-f003:**
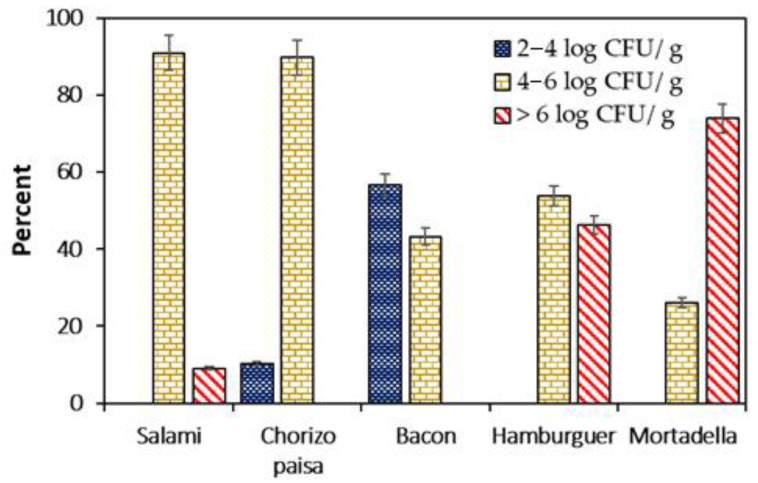
Abundance (percentages of CFU/g) of *Listeria monocytogenes* in the different meat products (Salami, “Chorizo paisa”, bacon, hamburguer and Mortadella) analyzed.

**Figure 4 foods-12-02956-f004:**
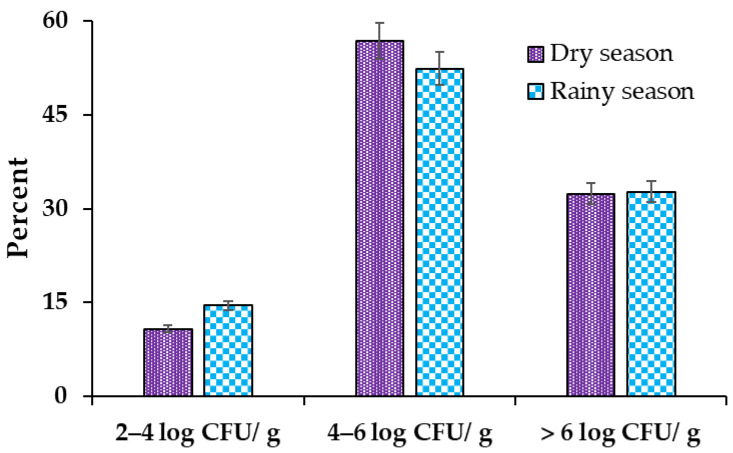
Abundance (percentages of CFU/g) of *L. monocytogenes* in the different meat products (Salami, “chorizo paisa”, bacon, hamburguer and mortadella) analyzed at different seasons. Marginal differences were found in the respective counts between the two seasons. In both seasons, 4-6 log CFU/g counts were found to be the highest, followed by >6 log CFU/g and 2-4 log CFU/g counts.

**Table 1 foods-12-02956-t001:** Meat product types, description, storage temperatures and geographical zones in Quevedo city, analyzed for the *L. monocytogenes* investigation.

Meat Product Type	Product Description	Storage Temperature		Geographical Zone (Number of Samples)		
		Dry Season	Rainy Season	South (%)	Central (%)	North (%)
A: Salami	Ripened product (200 g slices)	2–14 °C	4–14 °C	33	33	34
B: “Chorizo paisa”	Raw cured product, vacuum-packed presentation of 300 g			33	33	34
C: Bacon	Cured, smoked product, presented in 200 g slices			33	33	34
D: Hamburger meat	Roasted product, in 110 g package			33	33	34
E: Mortadella	Cooked product, 100 g presented in vacuum package.			33	33	34

**Table 2 foods-12-02956-t002:** Prevalence of *Listeria monocytogenes* in different types of meat products.

Types of the Meat Product	Number of Samples Collected in the Two Seasons	Number of Samples (Dry and Rainy Season) Positive for *L. monocytogenes*	Percentage of Positive Samples
A: Salami	200	21	10.5%
B: “Chorizo paisa”	200	29	14.5%
C: Bacon	200	30	15.0%
D: Hamburger meat	200	38	19.0%
E: Mortadella	200	45	22.5%

**Table 3 foods-12-02956-t003:** The prevalence (%) of *Listeria monocytogenes* in different seasons (dry–rainy) and geographical areas carried out in this study.

Type of Meat Product	Season				
	Dry (%)	Rainy (%)	South (%)	Central (%)	North (%)
A: Salami	12	9	39.3	40.5	20.2
B: “Chorizo paisa”	21	8			
C: Bacon	22	8			
D: Hamburger meat	30	8			
E: Mortadella	26	19			

## Data Availability

The data used to support the findings of this study have published in the tables and Figures of this work.

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
