# Peer review of "Prevalence of *Listeria monocytogenes* in RTE Meat Products of Quevedo (Ecuador)"

_foods, 2023, doi:10.3390/foods12152956_

Round 1

Reviewer 1 Report

The article concerns types of food common in the country of origin (Ecuador), with names that refer to international products (eg. Bologna mortadella) but a brief description of the same foods would be useful to make the type of food treated more understandable to international readers.

In the results they talk about positive samples for the presence of L. monocytogenes with values ​​in the count always higher than 4 log, but then in the discussion they make references to percentages of positivity lower than 2 log, values ​​never found by the authors.

 The type of graphs used to better illustrate the results do not help understanding. For example, it is suggested to use histograms and to always use the same measurement scale.

In the discussion reference is made to Reg 2073/2005 CE and s.m.a. and to the indications relating to “foods that support or do not support” the growth of L. monocytogenes based on the pH and aw values ​​of the food but in the work these parameters were not measured. It is therefore assumed that the foods under study fall into the category of “foods that support the growth of L. monocytogenes” and therefore in this case the criterion that should be applied it’s always presence/absence. The “100 cfu/g” criterion should in any case be evaluated at the end of the shelf life of the food, but according to the same indication of the authors, the foods were sampled far from the end of their commercial life and therefore the values ​​obtained as Lm counts are subject to a possible further increase considering that the same foods are not kept in the correct refrigeration condition.

It seems to me that the very worrying aspect determined by the high concentration levels of L monocytognes in the foods investigated is not sufficiently addressed in the conclusions and that no hypothesis is made on how action should be taken to improve the situation

Minor revisions
line 18: “foodborne pathogen, the causative” is missing “is“ after the comma.

Line 19-20: “temperature” instead of “temperatures”

Line 25: “performed” instead of “detected”

In all the paper “Bologna” must be written with an initial capital letter, not bologna

Line 45: change “study” with “several studies”

Line 70: “refrigerated conditions” : specify how many °C

Line 75 “close to expiration”: specify, how close?

Line 84: change “To do this” in “According to the method”

Line 89: add at the end “and are observed after 48hh of incubation of the plates seeded with the enrichments broths”

Line 96 Change “24h” in “24-48 h”

Line 115-116: 50 ul = 29 ul + 2 ul + 2 ul + 2 hl (35 ul). 15 ul are missing to make 50ul, explain better

Line 121-128: What’s the range of bands? 300-1000 or 100-1000? Explain better

Line 141: change “that were confirmed all of them by” in “that were all confirmed by”

Line 143: change “In all of them was amplified” in “In all of them it was amplified” and delete “approximately”

Line 144: change “700 pb” in “660 pb”

Figure 1 pag molecular weight: 200pb and 100pb are missing; markers are missing

Line 158-165: the period is not written in a very clear way and moreover it reports exactly what is written in Table 2, therefore it seems a useless repetition

Line 169: change “According to” in “taking into consideration”

All the papar sometimes they use “L. monocytogenes”, sometimes “Listeria monocytogenes” standardize and evaluate whether to abbreviate in “Lm

Table 3: Dry season % or Rainy season %: the numbers reported are not the percentage of samples but are the number of positive samples

Figure 3: change the type of graphs and put the name of the food under the specific graph and not in the legend

Line 207-217: the period is not written in a very clear way and moreover it reports exactly what is written in Figure 3, therefore it seems a useless repetition

Line 214: “53% “ it’s actually 53,8%

Line 214: “Higher than 4 log” it’s actually “between 4-6 log”

Line 215: “more of 46%” it’s actually 46,2%

Line 221 change “study” in “studied”

Line 225: change “times” in “periods”

Line 225-231: the period is not written in a very clear way and moreover it reports exactly what is written in Figure 4, therefore it seems a useless repetition

Table 4: where do the percentages of samples with count values < 2log come from? Review from line 237 to 251

Line: 253 change “the prevalence is” in “the prevalence was”

Line 254: delete “one of” and change “is” in “are”

Line 255: change “the raw material” in “the contamination of raw material”

Line 259: “preparation” is repeated twice

Line 260-266: this concept is not expressed very clearly

Line 286 change “higher those” in “higher than those; in which countries?

Line 297: Spanish legislation? Spain follow Eu Legislation (Reg 2073/2005)

The article is written in an English that is not always perfectly understandable.

Author Response

Reviewer 1.

Comment: The article concerns types of food common in the country of origin (Ecuador), with names that refer to international products (eg. Bologna mortadella) but a brief description of the same foods would be useful to make the type of food treated more understandable to international readers.

Response: Definitions of meat samples used according to the Mandatory Ecuadorian Technical Standard (NTE INEN 1338-2102) third revision 2012-04 are as follows: salami is a dry-cured fermented sausage made from pork and/or bovine meat, with pork fat, salt, sugar, spices with or without the addition of liquors; “chorizo Paisa” sausage is the product made stuffed in natural or artificial casings ripened for short period, smoked or not; bacon is the product obtained from the abdominal wall (bacon), or from the subcu-taneous adipose tissue of pigs, cooked and smoked; mortadella is an emulsified mass pre-pared with selected meat and fat from farm animals, ingredients and stuffed in natural or artificial casings cooked; roasted hamburger is the ground (or minced) meat of animals for sale, homogenized, preformed and precooked, and with ingredients and additives for permitted use. This information has been included in the revised version of the manuscript to give more details to the analyzed products to the readers.

Comment: In the results they talk about positive samples for the presence of L. monocytogenes with values in the count always higher than 4 log, but then in the discussion they make references to percentages of positivity lower than 2 log, values never found by the authors.

Response: In the discussion it was mentioned that in the present work  there were counts greater than 2 log, marked as non-compliance with the microbiological criteria for RTE meat products in the EU regulation. In addition, it is discussed that the levels of L. monocytogenes found in this work are higher than those indicated by the EFSA as possible risk of Listeriosis after consumption of contaminated RTE foods.

Comment: The type of graphs used to better illustrate the results do not help understanding. For example, it is suggested to use histograms and to always use the same measurement scale.

Response: As suggested the Reviewer 1, all the graphs were modified.

Comment: In the discussion reference is made to Reg 2073/2005 CE and s.m.a. and to the indications relating to “foods that support or do not support” the growth of L. monocytogenes based on the pH and aw values of the food but in the work these parameters were not measured. It is therefore assumed that the foods under study fall into the category of “foods that support the growth of L. monocytogenes” and therefore in this case the criterion that should be applied it’s always presence/absence. The “100 cfu/g” criterion should in any case be evaluated at the end of the shelf life of the food, but according to the same indication of the authors, the foods were sampled far from the end of their commercial life and therefore the values obtained as Lm counts are subject to a possible further increase considering that the same foods are not kept in the correct refrigeration condition.

Response: The samples of the meat products analyzed in the present work were taken in during the marketing at the middle of shelf life. Thus, although most of the analyzed products fall in the category of “foods that support the growth of L. monocytogenes”, as the sampling is in the marked the microbiological criterium applicated is the <100 CFU/g. The noticeable results are that a high percentage of samples show counts higher that the above microbiological criterium for L. monocytogenes. For this reason, is indicated in the conclusion of the revised manuscript that “effective temperature control throughout the chain are required. In addition, production, distribution and storage in the commercial refrigerators that guarantee the safety of meat products in supermarkets in Quevedo (Ecuador) to avoid risks of infection by foodborne listeriosis.”

Comment: It seems to me that the very worrying aspect determined by the high concentration levels of L monocytogenes in the foods investigated is not sufficiently addressed in the conclusions and that no hypothesis is made on how action should be taken to improve the situation

Response: Following details were included in the conclusion section. Sufficient programs should be implemented to prevent the contamination or inhibit the growth of L. monocytogenes in meat products, such as adequate cleaning, sanitation and hygienic programs and effective temperature control throughout the chain of production, distribution and storage in the commercial refrigerators that guarantee the safety and security of meat products in the supermarkets of Quevedo (Ecuador) to avoid risks of infection by food-borne listeriosis.

Minor revisions

line 18: “foodborne pathogen, the causative” is missing “is“ after the comma.

Response: This sentence has been modified.

Line 19-20: “temperature” instead of “temperatures”

Response: This change has been corrected in the new version of the manuscript.

Line 25: “performed” instead of “detected”

Response: The sentence was rewritten for its better understanding, and, in the new version of the manuscript, it no longer includes that word.

In all the paper “Bologna” must be written with an initial capital letter, not bologna.

Response:  The authors apologize for this error,the actual produce name is mortadella.

Line 45: change “study” with “several studies”

Response: Corrected accordingly.

Line 70: “refrigerated conditions”: specify how many °C

Response: Values of temperatures were included in this revised manuscript.

Line 75 “close to expiration”: specify, how close?

Response: The samples were collected at the middle of their half-life stage.

Line 84: change “To do this” in “According to the method”

Response: This change was made to the sentence.

Line 89: add at the end “and are observed after 48hh of incubation of the plates seeded with the enrichments broths”

Response: Sentence was added.

Line 96 Change “24h” in “24-48 h”

Response: Respective change was made.

Line 115-116: 50 ul = 29 ul + 2 ul + 2 ul + 2 µL  +15 µL

Response: The composition and respective volumes were corrected in the revised manuscript.

Line 121-128: What’s the range of bands? 300-1000 or 100-1000? Explain better.

Response: The range of the bands were 100 – 1000 and this was located at the weight of 660 base pairs corresponding to the molecular weight marker (ladder).

Line 141: change “that were confirmed all of them by” in “that were all confirmed by”

Response:  This sentence was rewritten in the new version on the manuscript.

Line 143: change “In all of them was amplified” in “In all of them it was amplified” and delete “approximately”

Response: This paragraph has been rewritten in the revised manuscript.

Line 144: change “700 pb” in “660 pb”

Response: This has been corrected.

Figure 1 pag molecular weight: 200pb and 100pb are missing; markers are missing.

Response: Markers were indicated in the figure.

Line 158-165: the period is not written in a very clear way and moreover it reports exactly what is written in Table 2, therefore it seems a useless repetition.

Response: Respective corrections were made in the Table 2.

Line 169: change “According to” in “taking into consideration”.

Response: This change has been doing.

All the papar sometimes they use “L. monocytogenes”, sometimes “Listeria monocytogenes” standardize and evaluate whether to abbreviate in “Lm

Response: In the whole manuscript, the name of the bacterium was indicated as “L. monocytogenes”; whereas at its first appearance in the abstract, figures, tables and titles, it was indicated in its full form as “Listeria monocytogenes”.

Table 3: Dry season % or Rainy season %: the numbers reported are not the percentage of samples but are the number of positive samples.

Response: In table 3, the values ​​that were expressed in the two seasons were the number of positive samples; and the geographic areas were in percentages

Figure 3: change the type of graphs and put the name of the food under the specific graph and not in the legend.

Response: As the Reviewer 1 suggested, the type of graphs has been changed and also, the names have been put in the graph.

Line 207-217: the period is not written in a very clear way and moreover it reports exactly what is written in Figure 3, therefore it seems a useless repetition.

Response: Text was corrected in figure.

Line 214: “53% “ it’s actually 53,8%

Response: This was amended in the new version on the manuscript.

Line 214: “Higher than 4 log” it’s actually “between 4-6 log”

Response:  Corrected.

Line 215: “more of 46%” it’s actually 46,2%

Response: Value digit was corrected.

Line 221 change “study” in “studied”

Response: Word corrected.

Line 225: change “times” in “periods”

Response: Word was changed.

Line 225-231: the period is not written in a very clear way and moreover it reports exactly what is written in Figure 4, therefore it seems a useless repetition.

Response: Corrections were made in the revised manuscript in order to avoid the repetition of the text.

Table 4: where do the percentages of samples with count values < 2log come from? Review from line 237 to 251.

Response: The percentages belong to the incidence of Listeria monocytogenes in the two seasons.

Line: 253 change “the prevalence is” in “the prevalence was”.

Response: This change has been made.

Line 254: delete “one of” and change “is” in “are”.

Response: These changes have been corrected accordingly.

Line 255: change “the raw material” in “the contamination of raw material”.

Response: Text was corrected.

Line 259: “preparation” is repeated twice.

Response: Repeated text was deleted.

Line 260-266: this concept is not expressed very clearly.

Response: Text was revised for a better clarity and readability.

Line 286 change “higher those” in “higher than those; in which countries?.

Response: Sentence was corrected.

Line 297: Spanish legislation? Spain follow Eu Legislation (Reg 2073/2005).

Response:  Spain follows EU legislation (Reg 2073/2005) as it is part of the European Union

The article is written in an English that is not always perfectly understandable.

Response: The whole manuscript was edited for the English language thoroughly word-by-word in this revised version.

Reviewer 2 Report

The manuscript is relevant; several corrections and careful review of the manuscript are required to make it publishable, including improved quality of English language.

Please see below the improvements / comments:

Line-18: please add the abbreviation “(L. monocytogenes)” after Listeria monocytogenes

Line-18: please add “is” before “the causative”

Line-19: please replace “and” with “that”

Lines-24-25: please replace “In the products where the presence of L. monocytogenes was detected” with “In the positive samples”

Line-25: please replace “this pathogenic bacterium” with “L. monocytogenes”

Line-25: please remove “total of”

Line-28: please remove bracket

Line-28: the data reported in this sentence no longer appear in the manuscript, why?

Line-29: please remove “to L. monocytogenes in all the analyzed products”

Line-30: please add “of L. monocytogenes ranging fromafter “levels”

Line-35: please put “Particularly” before RTE at the end of sentence

Line-36: please replace “constituting” with “allowing”

Line-37: please replace “pathogenic microorganisms that transmit food-borne diseases” with “foodborne pathogens”

Line-39: please add the abbreviation “(L. monocytogenes)” after Listeria monocytogenes

Line-46: “general” should be removed

Line-57: please replace “in culture media and by” to “by microbiological and”

Line-59: please replace “pathogenic bacterium” with “foodborne pathogen”

Line-63: please replace “pathogenic bacterium” with “L. monocytogenes”

Line-69: please replace “the different” with “two different”

Line-70: please put the comma before “in vacuum… e after conditions”

Line-72: please put in brackets “2=200”, at point B

Line-73: for point E the number of samples is missing

Section 2.1: in the section, it should be specified that the samples were also stored at abusive storage temperature (as shown in the table 1)

Line-82: please remove the “point” at the end of sentence and replace “method of” with “of”

Line-86: the sentence “Then samples were incubated in Listeria Fraser TM MEDIA (Titan Biotech, Ltd) broth for 24 h at 37 °C.” is unclear

Line-89: please provide putting the “point” at the end of sentence

Line-92: the sentence “which specifies a horizontal method for the enumeration 92 of L. monocytogenes” is superfluous

Line-93: please add “buffered” before “peptone”

Lines-94-95: it would be better to replace the sentence “Then the necessary dilutions were made, 0.1 mL of the dilutions to be analyzed” with “Then decimal dilutions were prepared and 0.1 mL of ach”

Line-97: “pulsed” is unclear, you can replace it with “presumptive”

Lines-104-105: the sentence “Extraction and preparation of the DNA template was performed from isolated bacterial cultures in enriched Listeria Fraser broth for 24 h at 37 °C.” is unclear

Line-109: change the sentence to “PCR assay”

Lines-110-120: please consider to arrange the whole section

Line-132: please replace “Listeria monocytogenes” with L. monocytogenes

Line-133: please put bracket after “mortadella”

Section 3.1: please consider to arrange the whole section, it gets confusing

Section 3.2: please consider to arrange the whole section, it gets confusing

Line-207: please remove “to L. monocytogenes”

Line-208: please replace “this pathogenic bacteria” with “L. monocytogenes

Line-211: please replace “between” with “between”

Line-242: please add the “point” at the end of sentence

Line-255: please remove the space before comma at the end of the sentence

Line-261: why do you indicate that “… temperatures…fluctuated between 5 °C and 18 °C” if then in the text there is 2-14 °C?

 Line-302: please remove the space before comma after products

Lines-310-312: please review the entire sentence

Line-316: please replace “Listeria monocytogenes” with L. monocytogenes

Line-320: please add “ranging fromafter “levels”

The quality of English needs to be improved

Author Response

Reviewer 2.

The manuscript is relevant; several corrections and careful review of the manuscript are required to make it publishable, including improved quality of English language.

Please see below the improvements / comments:

Line-18: please add the abbreviation “(L. monocytogenes)” after Listeria monocytogenes

Response: We express the name of the bacterium in an abbreviated form (i.e. L. monocytogenes) from its second appearance onwards in both sections i.e. Abstract and Main Text (from introduction section on wards). Whereas at its first appearance, we indicated it in full form (i.e. Listeria monocytogenes) as well as in titles, figures and tables.We believe that readers can follow it easily.

Line-18: please add “is” before “the causative”.

Response: This sentence has been modified in the new version.

Line-19: please replace “and” with “that”.

Response: The sentence has been rewritten for better understanding.

Lines-24-25: please replace “In the products where the presence of L. monocytogenes was detected” with “In the positive samples”.

Response: As Reviewer 2 suggested, this sentence was edited.

Line-25: please replace “this pathogenic bacterium” with “L. monocytogenes”.

Response: The sentence has been rewritten.

Line-25: please remove “total of”.

Response: This change has been done.

Line-28: please remove bracket.

Response: The bracket has been removed.

Line-28: the data reported in this sentence no longer appear in the manuscript, why?.

Response: The text was revised in the revised version.

Line-29: please remove “to L. monocytogenes in all the analyzed products”.

Response: The phrase was removed

Line-30: please add “of L. monocytogenes ranging fromafter “levels”.

Response: Text was corrected manuscript

Line-35: please put “Particularly” before RTE at the end of sentence.

Response: The word “Particularly” was added before RTE.

Line-36: please replace “constituting” with “allowing”.

Response: This change was made in the corrected version.

Line-37: please replace “pathogenic microorganisms that transmit food-borne diseases” with “foodborne pathogens”.

Response: As Reviewer 2 suggested, this change has been done.

Line-39: please add the abbreviation “(L. monocytogenes)” after Listeria monocytogenes.

Response:  The abbreviation was not added in the new version because we consider that it is redundant, therefore we consider that it is not necessary

Line-46: “general” should be removed.

Response: The word “general” has been removed.

Line-57: please replace “in culture media and by” to “by microbiological and”.

Response: The suggested text was replaced.

Line-59: please replace “pathogenic bacterium” with “foodborne pathogen”.

Response: This was corrected in the new version.

Line-63: please replace “pathogenic bacterium” with “L. monocytogenes”.

Response: This has been replaced.

Line-69: please replace “the different” with “two different”.

Response: This change has been done.

Line-70: please put the comma before “in vacuum… e after conditions”.

Response: Text was edited accordingly.

Line-72: please put in brackets “2=200”, at point B.

Response: Corrected accordingly.

Line-73: for point E the number of samples is missing.

Response: Sample details were provided in this revised version.

Section 2.1: in the section, it should be specified that the samples were also stored at abusive storage temperature (as shown in the table 1).

Response: This was amended.

Line-82: please remove the “point” at the end of sentence and replace “method of” with “of”.

Response: All these changes have been made in the new version.

Line-86: the sentence “Then samples were incubated in Listeria Fraser TM MEDIA (Titan Biotech, Ltd) broth for 24 h at 37 °C.” is unclear.

Response: The sentence was revised for a better clarity.

Line-89: please provide putting the “point” at the end of sentence.

Response: A point has been put at the end of the sentence.

Line-92: the sentence “which specifies a horizontal method for the enumeration 92 of L. monocytogenes” is superfluous.

Response: This sentence has been deleted.

Line-93: please add “buffered” before “peptone”.

Response: The word has been added.

Lines-94-95: it would be better to replace the sentence “Then the necessary dilutions were made, 0.1 mL of the dilutions to be analyzed” with “Then decimal dilutions were prepared and 0.1 mL of ach” .

Response: The Sentence was replaced as suggested.

Line-97: “pulsed” is unclear, you can replace it with “presumptive”.

Response: The word “pulsed” was replaced by “presumptive”.

Lines-104-105: the sentence “Extraction and preparation of the DNA template was performed from isolated bacterial cultures in enriched Listeria Fraser broth for 24 h at 37 °C.” is unclear.

Response: Sentence was corrected to a better understanding.

Line-109: change the sentence to “PCR assay”.

Response: Corrected

Lines-110-120: please consider to arrange the whole section .

Response: As Reviewer 2 suggested, whole section was restructured.

Line-132: please replace “Listeria monocytogenes” with L. monocytogenes” .

Response: This error has been corrected.

Line-133: please put bracket after “mortadella”.

Response: Corrected.

Section 3.1: please consider to arrange the whole section, it gets confusing.

Section 3.2: please consider to arrange the whole section, it gets confusing.

Response: Both sections were rearranged in the new version of the manuscript to make it easier to understand.

Line-207: please remove “to L. monocytogenes”.

Response: This paragraph has been rewritten.

Line-208: please replace “this pathogenic bacteria” with “L. monocytogenes”.

Response: This paragraph has been rewritten.

Line-211: please replace “beteween” with “between”.

Response: This paragraph has been rewritten.

Line-242: please add the “point” at the end of sentence.

Response: Corrected.

Line-255: please remove the space before comma at the end of the sentence.

Response: This was amended in the new version.

Line-261: why do you indicate that “… temperatures…fluctuated between 5 °C and 18 °C” if then in the text there is 2-14 °C?

Response: This error has been corrected.

Line-302: please remove the space before comma after products.

Response: The space was removed.

Lines-310-312: please review the entire sentence.

Response: The sentence was revised.

Line-316: please replace “Listeria monocytogenes” with L. monocytogenes”.

Response: This was amended in the new version.

Line-320: please add “ranging fromafter “levels”.

Response: The sentence was edited as the Reviewer 2 suggested.

Reviewer 3 Report

The work has the value of describing data that have not been collected so far in Ecuador, emphasising the need to reinforce hygienic conditions, and particularly temperature control throughout the process. The work obtains prevalence data on 6 products, differentiating two seasons and 3 zones in the region.

Abstract

Line 21. End inverted commas left over from bacon.

Keywords. I suggest not including primer (if at all, include the first concrete), and include the territorial aspect (Quevedo, Ecuador).
The paper describes the prevalence of Listeria monocytogenes in RTE products in the area of Quevedo (Ecuador).

In general, the background could be improved with more updated or more internationally relevant incidence and outbreak data. This could also enrich the discussion, especially by collecting generic and updated references on the influence of environmental factors on Listeria.

Reference 15 refers to a paper on viruses. Is right? (In relation to the sentence included in the background).

The first citation refers to "recent years", being a 2010 citation. This should be updated or modified.

Material and Methods

- Better to change "Taking samples" to "Sampling".
- Table 1. % of South, Center and North add up to 200, instead of 100. Please clarify.
- Line 87. Listeria must be in italics.
- Line 96. Listeria must be in italics.
- Line 133. Missing closing parenthesis.

Results

- Table 3. What is the meaning of the values in the columns Dry season (%) and Rainy season (%)? It should be expressed in the legend or in the table footnote.

Discussion

- Lines 253-256. The possible causes for the presence of Listeria are described in the reference. Why do the authors think that the results prove that? Perhaps they want to express that the occurrence in all products proves the wide distribution of Listeria along the whole food chain of these products? That would be more logical.

- Line 255. "Finished". Is there something missing in that expression?

Conclusion

"Hot season" has not been used as a reference so far.

English language must be thoroughly revised.

Author Response

Reviewer 3.

The work has the value of describing data that have not been collected so far in Ecuador, emphasising the need to reinforce hygienic conditions, and particularly temperature control throughout the process. The work obtains prevalence data on 6 products, differentiating two seasons and 3 zones in the region.

Response: The work presented data on the prevalence of L. monocyotogenes, studied in five meat products, two seasons of the year, samples collected in 3 areas of Quevedo.

Abstract

Line 21. End inverted commas left over from bacon.

Response: The inverted commas was deleted

Keywords. I suggest not including primer (if at all, include the first concrete), and include the territorial aspect (Quevedo, Ecuador).

Response: Keywords were revised and modified.

The paper describes the prevalence of Listeria monocytogenes in RTE products in the area of Quevedo (Ecuador).

In general, the background could be improved with more updated or more internationally relevant incidence and outbreak data. This could also enrich the discussion, especially by collecting generic and updated references on the influence of environmental factors on Listeria.

Reference 15 refers to a paper on viruses. Is right? (In relation to the sentence included in the background).

Response: Appropriate reference was included in this revised version.

The first citation refers to "recent years", being a 2010 citation. This should be updated or modified.

Response: Citation updated 2023

Material and Methods

- Better to change "Taking samples" to "Sampling".

Response: As suggested the Reviewer 3, “Taking samples” has been changed by “Sampling”.

- Table 1. % of South, Center and North add up to 200, instead of 100. Please clarify.

Response: The meat products were chosen from three points of sale that according to geographical location (such as north, south and central part); and in each of the points of sale, 200 samples of each meat product were chosen.

The values have been expressed in percentages in the new version of the manuscript.

- Line 87. Listeria must be in italics.

-Line 96. Listeria must be in italics.

Response: This was amended.

- Line 133. Missing closing parenthesis.

Response: The closing parenthesis has been added.

Results

- Table 3. What is the meaning of the values in the columns Dry season (%) and Rainy season (%)? It should be expressed in the legend or in the table footnote.

Response: In Table 3. We proceeded to place the response those requested by the reviewer, which is in the revised version of the manuscript.

Discussion

- Lines 253-256. The possible causes for the presence of Listeria are described in the reference. Why do the authors think that the results prove that? Perhaps they want to express that the occurrence in all products proves the wide distribution of Listeria along the whole food chain of these products? That would be more logical.

Response: It is more logical to express that “the occurrence of L. monocytogenes in all products that proves the wide distribution of this pathogenic bacteria along the whole food chain of these products”. We followed the same line in this revised version of manuscript.

- Line 255. "Finished". Is there something missing in that expression?

Response: Expression was amended in the revised version of the manuscript.

Conclusion

"Hot season" has not been used as a reference so far.

Response: Expression was corrected in the manuscript where ever necessary.

English language must be thoroughly revised.

Response: Whole manuscript was edited for English language thoroughly sentence-by-sentence without affecting the flow of information/contents in this revised version.

Round 2

Reviewer 1 Report

The revised article is now certainly better understood but the English should be revised to make further improvements.

Unfortunately it seems to me that the authors did not fully understand the indications given in Reg 2073 /05CE concerning the safety criterion referring to Listeria monocytogenes: Reg 2073/05 CE does not say that the safety criterion for RTE foods is < 100 cfu/g as the authors continue to claim.

This limit can be used as a reference only for those foods that do not support the growth of the microorganism, i.e. with specific pH, aw values, residual shelf life < 5 days or if the manufacturer has performed experimental studies that can demonstrate it. Otherwise and for the products presented in the article, only the safety criterion of absence of L. monocytogenes must be considered.

The considerations about the count of Lm reported in the results from Table 4 up to the end of the paragraph are devoid of scientific significance and in my opinion should be eliminated.

The statements reported in the paragraph of the discussion (lines 409 to 413) do not correspond with the indications of the Reg. 2073/05 CE

Minor revisions
Line 32 change “abundance” in “concentration”

Line 219: it’s written “aforementioned”, it’s supposed to be “before mentioned”

Line 379 correct “cooked hum” in “cooked ham”

Line 407 change “incidence” in “presence”

Line 411: “With the this results” delete “the”

Line 412 correct “consideret” in “considered”

See above

Author Response

Reviewer 1.

Comment: The revised article is now certainly better understood but the English should be revised to make further improvements.

Response: We thanks to the Reviewer 1 for this comment. The English has been again extensively revised in the new revised version of the manuscript to make further improvements.

Unfortunately it seems to me that the authors did not fully understand the indications given in Reg 2073 /05CE concerning the safety criterion referring to Listeria monocytogenes: Reg 2073/05 CE does not say that the safety criterion for RTE foods is < 100 cfu/g as the authors continue to claim. This limit can be used as a reference only for those foods that do not support the growth of the microorganism, i.e. with specific pH, aw values, residual shelf life < 5 days or if the manufacturer has performed experimental studies that can demonstrate it. Otherwise and for the products presented in the article, only the safety criterion of absence of L. monocytogenes must be considered.

Response: We agree with the Reviewer 1 about Regulation 2073 of European Commission. To avoid a misunderstanding the paragraph where this regulation is referred has been rewritten as fallow: “According to European Commission Regulation EC 2073/2005 the analyzed products are in the category of RTE foods able to support the growth of L. monocytogenes, thus, all of them have to comply the criteria of absence in 25 g before these foods has left the immediate control of the food business operator and less of 2 log CFU/g for these products placed on the market during their shelf-life. Thus, according with the above European Commission Regulation, non-compliance of criteria levels for L. monocytogenes were found in all types of the analyzed meat products in the present work”. In addition, more references to this regulation have been deleted in the revised version of the manuscript.

The considerations about the count of Lm reported in the results from Table 4 up to the end of the paragraph are devoid of scientific significance and in my opinion should be eliminated.

Response: This paragraph and table 4 have been eliminated from the revised version of the manuscript, since according to Reviewer 1 they lack scientific value.  

The statements reported in the paragraph of the discussion (lines 409 to 413) do not correspond with the indications of the Reg. 2073/05 CE

Response: This paragraph has been deleted and all paragraph in the Discussion section referring to Reg 2073/05 have been rewritten, as has been indicated previously.

Minor revisions
Line 32 change “abundance” in “concentration”

Line 219: it’s written “aforementioned”, it’s supposed to be “before mentioned”

Line 379 correct “cooked hum” in “cooked ham”

Line 407 change “incidence” in “presence”

Line 411: “With the this results” delete “the”

Line 412 correct “consideret” in “considered”

Response: all these minor changes have been corrected in the revised version of the manuscript.